# A Systematic Review of Breast Implant-Associated Squamous Cell Carcinoma

**DOI:** 10.3390/cancers15184516

**Published:** 2023-09-12

**Authors:** Sujan Niraula, Anjan Katel, Amit Barua, Anna Weiss, Myla S. Strawderman, Huina Zhang, Oscar Manrique, Avice O’Connell, Sirish Raj Pandey, Ajay Dhakal

**Affiliations:** 1Nuvance Health, Vassar Brother Medical Center, Poughkeepsie, NY 12601, USA; anjan.katel3@gmail.com (A.K.); amitbarua29@gmail.com (A.B.); sirishrajpandey@gmail.com (S.R.P.); 2Department of Surgery, University of Rochester, Rochester, NY 14642, USA; anna_weiss@urmc.rochester.edu (A.W.); oscar_manrique@urmc.rochester.edu (O.M.); 3Department of Biostatistics and Computational Biology, University of Rochester, Rochester, NY 14642, USA; myla_strawderman@urmc.rochester.edu; 4Department of Pathology, University of Rochester, Rochester, NY 14642, USA; 5Department of Imaging Sciences, University of Rochester, Rochester, NY 14642, USA; avice_oconnell@urmc.rochester.edu; 6Department of Medicine, University of Rochester, Rochester, NY 14642, USA; ajay_dhakal@urmc.rochester.edu

**Keywords:** breast implant-associated malignancies, squamous cell carcinoma, BIA-SCC, silicone implant, breast augmentation, mastectomy, breast swelling

## Abstract

**Simple Summary:**

Breast augmentation is a widely performed procedure worldwide and is considered safe. However, there is a concerning rise in rare cases of cancers associated with breast implants, including lymphoma and squamous cell carcinoma. Both the American Society of Plastic Surgeons (ASPS) and the FDA have issued warnings about these risks. Our comprehensive study of reported cases of breast implant associated squamous cell carcinoma (BIA-SCC) reveals that the treatment approaches for these conditions vary, and sadly, some patients have a short life expectancy after diagnosis. This emphasizes the critical need for enhanced monitoring and information sharing to detect and effectively manage BIA-SCC. Healthcare providers must maintain vigilance when conducting long-term follow-ups with patients who have undergone breast augmentation surgery.

**Abstract:**

Breast augmentation is considered safe, but rare cases of breast implant-associated squamous cell carcinoma (BIA-SCC) have been reported. This study aimed to systematically review published cases of BIA-SCC, providing valuable clinical data. The review included 14 articles and 18 cases of BIA-SCC. An increasing trend in reported BIA-SCC cases was observed, with four cases in the 1990s and 14 cases since 2010. The mean age of affected patients was 56 years, and symptoms typically appeared around 21 years after breast augmentation. Silicone implants used in cosmetic procedures were most commonly associated with BIA-SCC. Implant removal was necessary in all cases, and some patients required a mastectomy. Treatment approaches varied, with the selective use of chemotherapy and/or radiotherapy. The estimated 6-month mortality rate was 11.1%, while the 12-month mortality rate was 23.8%. The estimated 6-month mortality rate should be cautiously interpreted due to the limited sample size. It appears lower than the rate reported by the American Society of Plastic Surgeons, without clear reasons for this discrepancy. This study highlights the importance of enhanced monitoring and information sharing to improve detection and management of BIA-SCC. Healthcare providers should maintain vigilance during the long-term follow-up of breast augmentation patients.

## 1. Introduction

Breast augmentation is a widely utilized cosmetic procedure, with approximately 193,000 procedures in the United States alone in 2020 [1]. Despite being a standard procedure, it has associated complications, ranging from common complications like bleeding, infection, seroma, and hematoma formation to more severe outcomes such as skin necrosis, Mondor disease, capsular contraction, and implant rupture [2,3,4,5,6,7]. In addition, there is a risk of breast implant-associated (BIA) malignancies, such as various lymphomas including anaplastic large cell lymphoma (ALCL) and squamous cell carcinoma (SCC) [8]. In September 2022, the US Food and Drug Administration (FDA) issued a communication to raise awareness about the risk of developing BIA-SCC or BIA-ALCL in the scar tissue or capsule surrounding the implant, which was further revised in March 2023 [8,9,10,11]. BIA-SCC is a rare and poorly characterized disease with data limited to a few case reports. In response to the FDA’s communication, the American Society of Plastic Surgeons (ASPS) issued a statement regarding BIA-SCC based on 16 reported cases [10]. In this study, we aimed to systematically review the published cases of BIA-SCC and describe the relevant clinical and outcome data.

## 2. Materials and Methods

### 2.1. Search Strategy

We conducted a literature search on MEDLINE, EMBASE, and Google Scholar to identify studies reporting BIA-SCC from their inception to October 2022. Only studies conducted on humans were included, and the search was limited to the English language. We further identified relevant articles by screening the bibliography of the included articles using the snowball method.

### 2.2. Eligibility Criteria

Randomized controlled trials, prospective or retrospective case series, and case reports published in journals were eligible for our systematic review. We excluded review articles, discussions, and commentaries. Included studies had patients who underwent breast augmentation with implants and subsequently developed squamous cell cancer of the breast. Cases of breast augmentation after mastectomy for breast cancer that subsequently developed SCC were also included in our study. This systematic review study has not been registered in PROSPERO.

### 2.3. Study Selection

Reviewers (S.N. and A.K.) independently searched and screened the titles and abstracts for eligibility. Full texts of citations deemed eligible for our study were retrieved and further assessed for eligibility. Any disagreements were settled by a discussion between the reviewers. Relevant data were extracted by two authors (A.B. and A.K.) and checked for accuracy by S.N.; A.D. and A.W. participated in the final review.

### 2.4. Data Collection

We collected the patients’ baseline characteristics and relevant outcomes from individual studies, as reported in Table 1. In addition, the following data were extracted as available: age, the time interval from implant placement to SCC diagnosis, reported symptoms, type of implant used, imaging findings, prior history of breast cancer, surgical treatment, operative findings, presence of metastasis at the time of surgery, histology, hormone receptor status of the SCC (if assessed), adjuvant therapy, follow-up period, disease outcomes, and survival.

### 2.5. Data Synthesis

Aggregated data were collected from the included cases, and trends were described. The mortality rates at 6 and 12 months were calculated as the inverse of the Kaplan–Meier survival estimate and presented with a 95% confidence interval (CI).

## 3. Results

In this systematic review of published cases, we found 14 articles with 18 published cases of BIA-SCC. The demographics and baseline characteristics of the included cases are presented in Table 1.

### 3.1. Demographics, Nature of Implant, and Clinical Features

The patients had a mean age of 56 years (range: 40–81 years), and the mean time from breast augmentation to the appearance of signs or symptoms was 21 years (range: 10–35 years). Our findings showed that the mean age of the patients in the breast reconstruction arm was relatively older (65 years) than those in the cosmetic surgery arm (54 years). This is possibly due to cosmetic surgery being utilized more by the relatively younger population. The most common reason for breast augmentation was cosmetic (14/18, 78%). Six patients were triple negative; in the rest of the cases, receptors were not reported. Mutation carrier states were not analyzed as data were not available. Two patients received implant after mastectomy following breast cancer; one patient had implant after the removal of a benign mass. On average, the period from breast augmentation to cancer diagnosis was 21 years. Eleven patients had left-sided, six had right-sided BIA-SCC, and one had bilateral involvement. Sixteen patients had breast swelling, and fifteen had breast pain at the presentation of BIA-SCC. Two patients complained of discharge from the breast: one had a clear discharge, while the other had a purulent discharge [23,24]. Two patients reported a history of recent trauma to the breast [23]. The most common type of augmentation was silicone-based augmentation (ten silicone implants and two liquid silicone), with saline implants in five. All the other implants were capsulated except for the silicone injection. Among the five cases in which the surface of the implants was described, four patients had smooth and one had a textured implant.

### 3.2. Diagnosis, Assessment, and Histology

None of the patients had peri-implant effusions on breast imaging, of which three were suspected hematomas. The most commonly used diagnostic imaging modality was an ultrasound, followed by an MRI of the breast. The fluid was often white, thick, creamy, and cloudy. Operative findings varied among the cases and included findings like fluid accumulation (nine cases), mass (ten cases), cyst/nodule (seven cases), granulation (five cases), calcification (three cases), inflammation (three cases), and infection (two cases). Sentinel lymph node involvement was not seen in any cases. Among the 13 patients with mass size (cm) reported, the largest dimension was 4.9 cm (SD = 1.9), ranging from 1.5 to 8 cm. In the majority of the cases, growth was seen on the capsule lining in nest and bundles. Similarly, the cases with silicone injection had multiple cysts. SCC from primary other than breast capsules was not reported in any cases. Regarding the described morphology of the tumor, five tumors were well-differentiated, five were moderately differentiated, three were poorly differentiated, and one patient had variable differentiation of SCC. An undefined grade was present in one patient, and metaplasia was seen in three patients. The hormone receptor status was not reported in 12 cases, reported 6 cases were ER, PR and HER2 negative.

### 3.3. Treatment, Complication, and Prognosis

In our systematic review, a total of five cases reported capsular contraction. All patients had their implants removed, and 11 underwent a mastectomy. Of the 12 cases with available treatment information, seven received adjuvant chemotherapy and/or radiotherapy, two received neoadjuvant chemotherapy, and three did not receive perioperative chemotherapy or radiotherapy. Among the reported cases, two patients were lost to follow-up, four patients had remission, and two did not have significant progression at the time of reporting. Four had disease progression, of which three subsequently died. One patient received palliative radiation due to locoregional metastasis [12]. Among the patients who died, one had significant disease progression without receiving chemotherapy [12] and another patient developed malignant pleural effusion due to the invading mass and died after three months of diagnosis while still being on chemotherapy [23]. The last patient died after one year of diagnosis with significant disease progression [18]. Follow-up data were available for 10 cases (Table 2). The 6-month and 12-month mortality rates were 11.1% (95% CI: 0.5% to 40.6%) and 23.8% (95% CI: 2.8% to 56.1%). Notably, recently published reviews by Glasberg et al. and Möllhoff et al. did not provide any survival analysis [26,27].

Cases reported by Olsen et al. and Soni et al. received chemoradiation, but the name of the regimen was not mentioned [12,22]. The patient reported by Soni et al. was pregnant and underwent capsulectomy, implant removal, and mastectomy during her pregnancy, but chemotherapy and radiation were given after the pregnancy [14]. Four cycles of epirubicin, cyclophosphamide therapy, and four cycles of docetaxel therapy were used by Toyonaka et al. [21]. Liu et al. reported using gemcitabine combined with carboplatin followed by radiation therapy with the subsequent addition of capecitabine. However, the tumor metastasized under the left pectoralis major muscle near the axilla and at the left supraclavicular lymph nodes. The patient also had a history of ductal carcinoma in situ (DCIS) 10 years prior [20]. Both patients reported by Goldberg et al. were treated with fluorouracil and cisplatin [23].

## 4. Discussion

BIA-SCC is a rare and poorly understood disease with a poor prognosis. Data on BIA-SCC are limited, with a limited understanding of the pathophysiology of the disease [15]. It has been postulated that chronic foreign body irritation by the implant leads to chronic inflammation of the fibrous capsule and lymphatic system, resulting in chronic antigenic stimulation. Micro leaks from the implant may also contribute to chronic T-cell activation [14,28]. Additionally, a patient’s genetics and the growth of bacterial biofilm are also considered possible causes [29,30]. An alternative hypothesis suggests that microscopic fragments of epithelial tissue may proliferate into the lining of the implant during placement, leading to metaplasia and eventually transforming into malignancy [13]. New capsular epithelialization was initially considered benign and thought to serve as a precursor to squamous cell carcinoma by pathologists. Monitoring such cases could aid in earlier BIA-SCC detection and treatment [27]. None of these hypotheses have been scientifically investigated or established.

In this systematic review, we analyzed published cases of BIA-SCC and described the clinical and outcome information. Breast augmentation is a widely performed procedure, with an estimated 5 million to 35 million women worldwide having received breast implants. In 2020, 193,703 breast augmentations were performed in the United States, with 84% using silicone and 16% using saline implants. Silicone-based implants are the most commonly used type globally [1,9,31].

In our review, we found that breast enlargement was the most common presenting symptom of BIA-SCC, with a mean time from breast augmentation to diagnosis of 20 years. This highlights the importance of carefully evaluating late-presenting breast enlargement or seroma after breast implantation. The establishment of a new reporting system by the ASPS is expected to result in the increased reporting of cases and improved understanding of presentation and incidence [32]. In the meantime, the ASPS recommends that patients undergo MRI or ultrasound scans after 5–6 years of silicone implantation and every 2–3 years thereafter to screen for implant integrity and to rule out leakage or rupture as well as fluid surrounding the implant. Follow-up is crucial, as some patients may remain asymptomatic even after implant rupture [32,33].

The National Comprehensive Cancer Network (NCCN) has issued guidelines for the management of BIA-ALCL [34]. However, no such guidelines are available for BIA-SCC. Due to the increasing number of reported cases, the ASPS has outlined a rough guideline and recommendation regarding BIA-SCC. Patients with breast implant presenting with late-onset breast tenderness or enlargement should be evaluated for the possibility of BIA-SCC by ultrasound and/or MRI of the breast. If there is a mass around the implant, then the patient should undergo biopsy of the mass. If there is a seroma around the implant, fine-needle aspiration (FNA) followed by cytological evaluation and culture to differentiate between malignancy and infection should be performed. If morphologically suspicious, immunohistochemical stains for CD30, ALK, CK5/6, p63 assessment and flow cytometry for T-cells’ and B-cells’, squamous cell, and cytokeratin should be performed to rule out squamous cell carcinoma and lymphoma. Ultrasound and MRI with and without contrast should be performed. In confirmed cases, a PET/CT scan should be conducted before surgery to exclude distant metastasis. MRI and PET/CT scans offer heightened sensitivity compared to CT scans and ultrasound for evaluating patients and gauging the extent of their condition. Notably, a significant number of patients displayed extracapsular spread during an initial surgery, necessitating subsequent procedures. Comprehensive disease assessment from MRI could have influenced the surgical strategy. A tumor’s resistance to chemotherapy and radiation highlights the potential advantages of surgical resection. Additionally, after surgery, a post-surgical pathological assessment is necessary to distinguish between SCC and lymphomas. As per the ASPS guidelines, optimal outcomes are associated with total capsulectomy, while chemoradiation does not seem to yield benefits [10,11,27,35]. It is advisable to subject suspicious abnormalities discovered in breast implant capsules during routine implant exchange procedures to histologic analysis [11]. Of note, the ASPS recommendation is based on 16 cases and should be adopted very cautiously.

The ASPS has not included liquid silicone injection into consideration. However, in our systematic review, we also included cases that had augmentation with silicone injection. Although the clinical presentations of BIA-SCC with both silicone injection and implant were similar, perioperative findings showed multiple small cyst/nodules in silicone injection patients, whereas a mass on the capsule lining was seen in implant patients. BIA-ALCL has an increased incidence with textured implants [36]. Liquid silicone injection may not have been included by the ASPS as the FDA has never approved its use and banned its use in the early 1990s due to its adverse effects like granuloma formation, ulceration, fistula, discoloration, and the movement of silicon to other parts of the body [37]. Our study showed that BIA-SCC can occur with both smooth and textured implants. The histological findings of the deceased patients in various cases (10, 13, 15) showed moderate differentiation of squamous cell carcinoma [12,18,23]. However, due to the rarity of BIA-SCC and the limited number of reported cases, it remains uncertain whether the histological pattern is associated with mortality.

The ASPS reports a 6-month mortality rate of 43.8% for BIA-SCC [10]. Our calculations showed a 6-month mortality rate of 11% (95% CI: 0.5–40.6%). The reason for this difference in the estimate is unclear. The mortality rates in our study were calculated as the inverse of the Kaplan–Meier survival estimate from 10 cases with survival data reported. These survival data are clearly listed in Table 1. All these 10 cases are referenced in the ASPS statement as well, so they were likely included in the mortality estimate by the ASPS. The ASPS statement does not comment on its methodology for mortality estimation. Nevertheless, due to the small sample size, it is important to interpret these findings from both studies cautiously.

## 5. Conclusions

BIA-SCC is a rare and serious disease with limited diagnostic, management, and prognosis information. This entity has recently been recognized by the FDA and the ASPS. The true incidence of BIA-SCC is thought to be underreported, as patients and providers may not be aware of the condition unless the patient presents with significant signs or symptoms. Although the total number of reported cases of BIA-SCC is still low, there has been an increasing trend over the past several years, with 4 cases reported in the 1990s and 14 cases reported since 2010. Given the evolving knowledge of BIA-SCC, a comprehensive and well-established guideline for its diagnosis and management is currently absent, leading to the possibility of underdiagnosis, misdiagnosis, and improper treatment. It is crucial that healthcare providers and patients be aware of this serious condition. Continuous monitoring and sharing of information about this disease are crucial for advancing our knowledge and improving its detection and management. The findings from our study highlight the necessity for large national organizations such as the NCCN to develop comprehensive guidelines and recommendations for BIA-SCC.

## Figures and Tables

**Table 1 cancers-15-04516-t001:** Patient characteristics.

Author	Age (Years)	Time to Presentation (Years)	Clinical Features at Presentation	Type of Implant	Imaging	Surgical Treatment	Operative Findings	Metastasis	Histology	Adjuvant Therapy	Follow-up	Outcome	Survival
Olsen et al., 2017 [12]	56	28	Breast pain, enlargement, and skin discoloration	Silicone replaced by textured saline implants	NA	Surgical removal of both implants followed by left mastectomy	Thick, white fluid on incising the left breast capsule. Mass present on posterior surface.	Negative	Invasive, well- to moderately differentiated SCC	Chemotherapy and radiotherapy	Locoregional metastasis within 8 months	Treated with palliative radiation therapy due to metastasis	Alive at 1 year
	81	NA	Breast pain and enlargement	Silicone	US: partially cystic 2.9 cm breast mass suggestive of hematoma	Initially conservative then implant removal followed by left mastectomy for invasive SCC	NA	Negative	Invasive, moderately differentiated SCC	Chemotherapy and radiotherapy	Metastasis to liver and lung	Death	Died of disease in 1 year
Buchanan et al., 2018 [13]	65	35	Breast enlargement, tenderness	Foam-covered Silastic implants	Mammogram: edema vs. hemorrhage around left breast implant with superior extravasation of silicone material; US: circumferential hypoechogenicity edema vs. hemorrhage;	Complete capsulectomy with implant exchange followed by left radical mastectomy with medial chest wall resection	Periprosthetic milky fluid	Axillary lymph nodes and internal mammary lymph node chain	Well-differentiated SCC	Radiotherapy	After 8 years, disease-free	Disease-free	Alive at 8 years
Alikhan et al., 2010 [14]	70	16	Change in shape and size of breast	Silicone implant	MRI: ruptured implant with fluid and debris between the capsule and implant shell. An abnormal contour of the inferior margin.	Right breast capsulectomy, implant exchange, a Ryan flap, and right pocket revision	Creamy, white discharge	NA	Keratinizing squamous metaplasia with silicone debris and foreign body giant cell reaction	NA	NA	NA	Not reported
Kitchen et al., 1994 [15]	42	11	Breast pain on the first visit. After implant removal, presented with left breast mass.	Silicone implant	Mammogram after removal: bilateral large silicone granulomas	In 1991, implants were removed. In 1992, soft tissue mass was removed from the right breast.	Cyst contained thin, opaque, brown-yellow fluid.	NA	SCC	NA	NA	NA	Not reported
	52	15	Brest pain and enlargement	Silicone implant	NA	Surgical exploration initially, followed by left modified radical mastectomy	Intact and grossly unremarkable	NA	Poorly differentiated SCC	NA	NA	NA	Not reported
**Author**	**Age (Years)**	**Time to Presentation (Years)**	**Clinical Features at Presentation**	**Type of Implant**	**Imaging**	**Surgical Treatment**	**Operative Findings**	**Metastasis**	**Histology**	**Adjuvant Therapy**	**Follow-up**	**Outcome**	**Survival**
Paletta et al., 1992 [16]	52	15	Breast pain and enlargement	Silicone implant	NA	Surgical exploration followed by radical mastectomy	Sebaceous-type mass, long posterior capsule of the left implant. It had appearance of ruptured sebaceous cyst.	No	Focally invasive, variably differentiated SCC	Not given	Disease-free at 12 months of follow-up	Disease-free	Alive at 12 months
Talmor et al., 1995 [17]	70	25	Breast pain and enlargement	Silicone injection	Mammogram: large mass replacing entire breast; MRI: large fluid-filled cyst in left breast; US: markedly irregular architecture in both breasts	Bilateral simple mastectomy	Large, fluid-filled cyst in the left breast, multiple irregularly shaped silicone-filled cysts and nodules on both breasts.	No	Infiltrating, moderately differentiated SCC	NA	NA	NA	Not reported
Zhou et al., 2018 [18]	46	21	Breast pain and enlargement	Silicone implant	MRI: large fluid collection around the right implant	Surgical drainage of fluid and capsulectomy followed by bilateral capsulectomy. Later, underwent re-excision of remaining chest wall.	NA	Locoregional	Moderately differentiated SCC	Radiotherapy	Progressive metastasis to lung, liver, kidney, retroperitoneum, and right psoas.	Death	Died 17 months after diagnosis
Zomerlei et al., 2015 [19]	58	NA	Breast pain, enlargement, and edema	Silicone implant	NA	Fluid aspiration with keratinous debris drainage and implant removal. Right total mastectomy later.	Fungating mass on the posterior aspect of the right capsule was present.	Regional wall	Well-differentiated SCC	NA	NA	NA	Not reported
Liu et al., 2021 [20]	45	10	Breast enlargement and swelling	Silicone implant	Imaging suggesting metastasis to supraclavicular lymph nodes	Left chest wall mass resection, prosthesis removal, and left supraclavicular lymph node biopsy	NA	Yes	Poorly differentiated SCC	Chemotherapy and radiotherapy	Currently on OFS with oral anastrozole	No significant progress	Alive at 28 months from diagnosis
**Author**	**Age (Years)**	**Time to Presentation (Years)**	**Clinical Features at Presentation**	**Type of Implant**	**Imaging**	**Surgical Treatment**	**Operative Findings**	**Metastasis**	**Histology**	**Adjuvant Therapy**	**Follow-up**	**Outcome**	**Survival**
Toyonaka et al., 2022 [21]	51	16	Breast pain, swelling, and redness	Liquid silicone	US: fluid retention in left breast and enlarged reactive lymph nodes in left axilla; MRI: large ulcerative lesion in left breast. Irregularly shaped enhancing mass on base of the ulcer.	Partial left mastectomy followed by total left mastectomy and additional sentinel node biopsy	NA	NA	Well-differentiated SCC	Chemotherapy	NA	NA	Not reported
Soni et al., 2022 [22]	46	NA	Breast pain and swelling	Saline implants	NA	Modified radical mastectomy with en bloc excision of the implant and capsule	Opaque, tan periprosthetic fluid collection with pasty, white debris	No	Well-differentiated SCC	Chemotherapy and radiotherapy after pregnancy	In remission after 12 months of follow-up	Remission	Alive at 12 months
Goldberg et al., 2021 [23]	40	11	Swelling and erythema of left breast following breast trauma. Clear discharge from breast present.	Saline implants	CT: periprosthetic fluid with inflammation	Initially, bilateral implant removal with capsulectomies followed by further exploration 4 weeks later and was planned for chest wall resection	Intact implants with surrounding inflammation	Regional wall	Moderately differentiated infiltrating keratinizing SCC	Neoadjuvant chemotherapy before chest wall resection	Developed malignant pleural effusion secondary to invading mass	Death after 3 months of diagnosis	Died in 3 months
	62	32	Breast pain and enlargement after falling on her chest	Silicone implant	US: hematoma in right breast	Initially, bilateral implant removal and was planned for surgical resection after completion of neoadjuvant chemotherapy and radiotherapy	Small amount of turbid fluid on right breast with substantial granulomatous material and calcifications.	No	Well-differentiated invasive keratinizing SCC	Neoadjuvant chemotherapy before chest wall resection	Patient declined surgical resection and was offered palliative care	Lost to follow-up	Alive at 5 weeks
Whaley et al., 2022 [24]	60	26	Breast pain and enlargement. Skin color changes.	Saline implants	Large fluid collection surrounding left implant	Bilateral breast implant removal and capsulectomy	Purulent fluid and tan, verrucous proliferation along the inner lining of the left capsule.	No	Squamous metaplasia	Not given	No evidence of disease at 9 months of follow-up	No evidence of disease	Alive at 9 months
	57	25	Breast pain and enlargement on the first visit. On the second visit, presented with an open wound with yellow-green fluid draining from the breast.	Saline implants	US: complex hypoechoic fluid collection. Prominent intramammary and axillary lymph nodes. MRI: peripherally enhancing asymmetric mass with irregular borders and enlarged intramammary lymph nodes.	Capsulectomy and excision of wound edges	Poorly circumscribed, white-tan nodular proliferation along the breast capsule	NA	Squamous metaplasia	Not given	Lost to follow-up	Lost to follow-up	Not reported
Satgunaseelan et al., 2015 [25]	58	29	Breast pain and induration	NA	NA	Mastectomy	NA	NA	Squamous cell carcinoma high grade	NA	NA	NA	NA

**Table 2 cancers-15-04516-t002:** Vital status for 10 case reports.

Vital Status at Last Follow-up	Number of Patients	Follow-up Duration (mo.)	Reference
Alive	1	96	Buchanan et al. [13]
	1	28	Liu et al. [20]
	3	12	Paletta et al. [16] Olsen et al. [12] Soni et al. [22]
	1	9	Whaley et al. [24]
	1	1.25	Goldberg et al. [23]
Death due to BIA-SCC	1	17	Olsen et al. [12]
	1	12	Zhou et al. [18]
	1	3	Goldberg et al. [23]

## Data Availability

All data generated or analyzed during this study are included in this published article.

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
