# Peer review of "A Systematic Review of Breast Implant-Associated Squamous Cell Carcinoma"

_cancers, 2023, doi:10.3390/cancers15184516_

Round 1

Reviewer 1 Report

well written review, please emphazise in a subheadline/paragraph in discussion to instruct the clinicians what we should do to detect a BIA-SCC as early as possible. Is a late seroma aspiration the only way to diagnose the condition. It should be instructive for readers  

I doubt that 10 (!) authors contributed to this review without own cases. 

Reviewer 2 Report

Dear Editor and Authors,

Thank you for the opportunity to review the manuscript entitled “A Systematic Review of Breast Implant-Associated Squamous Cell Carcinoma”. The authors aimed to systematically review published cases of BIA-SCC, providing valuable clinical data. They included 14 articles and 18 cases of BIA-SCC. I am not aware of any similar systematic review concerning this very up to date and important topic. They concluded that his study highlights the importance of enhanced monitoring and information sharing to improve detection and management of BIA-SCC. Healthcare providers should maintain vigilance during long-term follow-up of breast augmentation patients. The conclusions are justified by the literature analysis and show the need for further studies on this topic. In recent months some articles concerning BIA-SCC appeared in the literature, and although the number of cases available may question the idea of systemic review, the paper comprehensively presents and summarized the current state of knowledge on the topic, so I consider it worth publishing in this Journal. 

However, I have some remarks:

-       There is a number on articles, communications in the available literature concerning the topic and, although not appropriate to be included in the systematic review, they should be mentioned in the intro or discussion, the authors should comment on these references and highlight their strengths in relation to those, e.g.:

Glasberg SB, Sommers CA, McClure GT. Breast Implant-associated Squamous Cell Carcinoma: Initial Review and Early Recommendations. Plast Reconstr Surg Glob Open. 2023 Jun 14;11(6):e5072. doi: 10.1097/GOX.0000000000005072.

-       This must be mentioned: Jewell ML, Walden JL, Fontbona M, Triana L. US FDA Safety Communication on Breast Implant Associated Squamous Cell Carcinoma BIA-SCC). Aesthetic Plast Surg. 2023 Apr;47(2):892-893. doi: 10.1007/s00266-023-03283-5.

-       What about this reference? Möllhoff N, Ehrl D, Fuchs B, Frank K, Alt V, Mayr D, Braig D, Giunta RE, Hagen C. Brustimplantat assoziiertes Plattenepithelkarzinom (BIA-SCC) – eine systematische Literaturübersicht [Breast implant-associated squamous cell carcinoma: a systematic literature review]. Handchir Mikrochir Plast Chir. 2023 Jul 20. German. doi: 10.1055/a-2108-9111. Epub ahead of print. PMID: 37473774. – this article is in german, which limits its availability, but you need to mention this!

-       Abstract: “Breast augmentation is generally safe” – “generally”? 

-       Introduction – FDA issued a report about BIA-SCC and various lymphomas (other than BIA-ALCL) not only BIA-ALCL /this was earlier/

-       I am lost with the design of tables /is there one or two/three tables?/ - this is just a matter of its graphical aspect /I do not have any comment regarding the data included/

-       “Nature of implant”? – type?

-       3.2 – you stated “the largest was…4.9 cm (SD=1.9), ranging from 1.5 to 8” – “1.5 to 8” what? cm?

-       Table 2/3? – vitat status – were these deaths related to SCC? /a comment to the table should be included/

Reviewer 3 Report

The title is "A systematic review of breast-implant associated squamous cell carcinoma".  It is mandatory to clarify if Authors analyzed breast cancer patients/ mutation carriers and implant based-breast reconstruction or healthy women after breast augmentation due to esthetic reasons.

Round 2

Reviewer 3 Report

Thank you for your clarification